# Towards Irreversible Attack: Fooling Scene Text Recognition via Multi-Population Coevolution Search

**Jingyu Li[1], Pengwen Dai[1]\*, Mingqing Zhu[1], Chengwei Wang[1], Haolong Liu[1], Xiaochun Cao[1]**
[1]School of Cyber Science and Technology,
Shenzhen Campus of Sun Yat-sen University, Shenzhen, Guangdong 518107, China
`{lijy768, zhumq8, liuhlong7}@mail2.sysu.edu.cn`,
`wangchw23@alumni.sysu.edu.cn`,
`{daipw, caoxiaochun}@mail.sysu.edu.cn`

## Abstract

Recent work has shown that scene text recognition (STR) models are vulnerable to adversarial examples. Different from non-sequential vision tasks, the output sequence of STR models contains rich information. However, existing adversarial attacks against STR models can only lead to a few incorrect characters in the predicted text. These attack results still carry partial information about the original prediction and could be easily corrected by an external dictionary or a language model. Therefore, we propose the Multi-Population Coevolution Search (MPCS) method to attack each character in the image. We first decompose the global optimization objective into sub-objectives to solve the attack pixel concentration problem existing in previous attack methods. While this distributed optimization paradigm brings a new joint perturbation shift problem, we propose a novel coevolution energy function to solve it. Experiments on recent STR models show the superiority of our method. The code is available at `https://github.com/Lee-Jingyu/MPCS`.

## 1 Introduction

Scene text recognition (STR) [29, 20, 30] has garnered increasing attention in the field of computer vision due to its wide applications, such as augmented reality [27], visual question answering [31], automatic driving [8], etc. Despite its great success, recent studies [32, 36, 37] have shown that STR models are vulnerable to adversarial examples, which deceive the models into making incorrect predictions by adding imperceptible perturbations on the input images. In practice, adversarial examples can be used to prevent private text [4, 37] in images from being recognized and exploited by malicious optical character recognition (OCR) systems. Therefore, adversarial attacks against STR models have become a valuable research topic.

Existing adversarial attacks in the computer vision field mainly focus on non-sequential tasks [9, 23, 33], such as image classification, object detection, face recognition, etc. The output of these models is usually a single label and contains limited information. In this scenario, once the model outputs a different label, the information conveyed by the output is completely altered. In contrast, the prediction of STR models can be seen as a series of character labels, which contain rich information. When attacking STR models, the altered prediction may retain partial information about the original prediction. With the retained information, the opponent could correct the mispredicted text through an extra dictionary [22, 30, 13] or large language models (LLMs) [34, 21, 15], as shown in Figure 1.

However, existing adversarial attacks against STR models consider merely altering the model prediction as a successful attack. For example, some works focus on attacking the text image with a clean

---

\*Corresponding author: Pengwen Dai `<daipw@mail.sysu.edu.cn>`

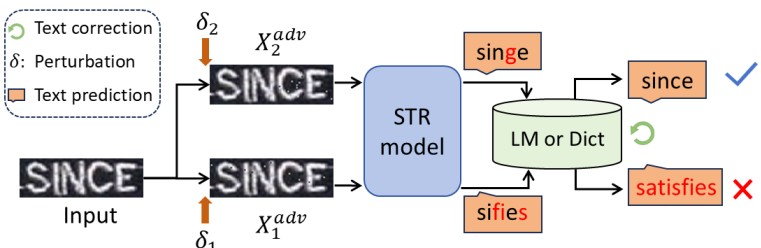

Figure 1: Adversarial attack results with a single incorrect character could be corrected through a language model or a dictionary. The attack results become more irreversible as the number of perturbed characters increases.

background [4, 11]. These methods may change the background or fail to attack due to complex backgrounds when dealing with scene text images. Differently, RoLMA [38] attacks license plate images, whose backgrounds are more complicated, by adding some light spots on the background. These light spots will also destroy the visual continuity of the background when dealing with scene text images. Compared to license plate images, scene text images are harder to attack due to their more complex backgrounds and diverse text styles. Xu et al. [36] develop an efficient optimization-based white-box attack method to fool STR models. Furthermore, in the black-box setting, $AD^2E$ [37] narrows the search space in the differential evolution algorithm to attack STR models. These methods overlook the fact that the attack results still contain rich information about the correct text.

In this work, we propose a black-box pixel-level attack method aiming at attacking all characters in the scene text image. We first discuss the reason for previous pixel-level attacks failing to attack more characters, *i.e.,* the attack pixel spatial concentration problem in their global optimization. Then we decompose the global optimization objective into several sub-objectives and introduce spatial constraints to avoid the attack pixel concentration. However, optimizing sub-objectives may undermine the global feasibility, manifested as the joint perturbation shift problem. Therefore, we propose the Multi-Population Coevolution Search (MPCS) method, which utilizes multiple evolutionary adversarial pixel populations in different image subspaces to individually search for each attack pixel in parallel. To solve the joint perturbation shift problem, MPCS introduces a coevolution energy function that implicitly embeds the global constraint into the local objective function. Experimental results on recent STR models demonstrate that MPCS produces substantially more incorrect predicted characters than state-of-the-art attack methods.

## 2 Related works

### 2.1 Attacks on common vision models

Existing adversarial attack methods mainly focus on non-sequential vision tasks, such as image classification, object detection, and facial recognition. These adversarial attack methods can be roughly categorized into white-box attacks and black-box attacks. White-box attacks use the whole information of the victim model, including model structure, parameters, and gradients. Fast gradient sign method (FGSM) [9] is proposed to generate adversarial perturbation according to the sign of the gradient. Deepfool [23] proposes to find the nearest classification hyperplane of the victim model to generate adversarial examples. GAT [12] composes multiple attack methods and utilizes a doubly stochastic matrix to optimize the attack order.

Differently, black-box attacks can only obtain the output of the victim model. OnePixel [33] employs the Differential Evolution (DE) algorithm to find a few adversarial pixels to attack image classification models. Transfer-based attacks [25, 1] utilize adversarial examples generated from attacking a source model to fool the target victim model. PRGF [5] introduces two prior-guided random gradient-free algorithms aimed at enhancing the efficiency of black-box adversarial attacks, which leverage a transfer-based prior provided by the gradient of a surrogate model. Furthermore, to enhance the transferability of adversarial perturbations, LLTA [7] generates generalized adversarial perturbations through data and model augmentation, and optimizes the final perturbation update through meta-learning.

These methods aim at attacking models that output non-sequential results, such as categories, positions, and scores. However, text recognition models predict text sequences, which usually contain more information for attackers to manipulate. We need to alter more predicted characters to ensure irreversible attack results.

## 2.2 Attacks on text recognition models

Text recognition models output text sequences containing more information than non-sequential outputs, which makes adversarial attacks aimed at them more challenging. [32] introduces an adversarial attack method targeting OCR systems, which involves making minor modifications to printed text images to deceive the OCR system into recognizing words with meanings opposite to the original text. [35] explores the effectiveness of the transfer-based attack algorithm in attacking real-world OCR systems. Another approach [4] involves manipulating the background of text images by replacing clean image backgrounds with random textures to disrupt the recognition by OCR systems. RoLMA [38] proposes a targeted attack method to fool license plate recognition models by adding some crafted light spots to images. Furthermore, Xu et al. [36] propose an optimization-based method to update the perturbation iteratively, attacking both CTC-based (connectionist temporal classification) and attention-based scene text recognition models. Based on the DE algorithm, $AD^2E$ [37] proposes to narrow down the continuous searching space to a discrete space, which can speed up the searching process. It generates adversarial examples by fixing a few pixels' values to 0 or 255 in the image.

These methods focus on changing the model output, which means attackers only need to change one character in the model predictions. However, a text with a few incorrect characters is easy to restore via the correction of an external dictionary or LLMs [34, 21, 15]. Therefore, in this work, we aim to attack all characters in images to make text recognition models predict more incorrect characters.

# 3 Preliminary

## 3.1 Problem formulation

Given a scene text image $X \in \mathbb{R}^{h \times w \times 3}$ ($h$ and $w$ are the height and width of the image), and its ground truth label $Y = [y_1, y_2, ..., y_n]$ ($n$ means the character number), the existing paradigm of crafting an adversarial example $X^{adv} = X + \delta$ is to add a small perturbation $\delta$ to $X$ to change the output of the victim model $\mathcal{F}$, such that,

$$\mathcal{F}(X) \neq \mathcal{F}(X + \delta), \quad \text{s.t.} \quad \|\delta\|_p \leq \epsilon, \tag{1}$$

where $\epsilon$ represents the maximum perturbation.

However, when attacking scene text recognition models, merely altering the output is not enough. We need to mislead the victim model to predict as many erroneous characters as possible, rendering the attack results more irreversible. So we formulate a new objective as follows,

$$\delta^* = \arg\max_{\delta} D\left(\mathcal{F}(X), \mathcal{F}(X + \delta)\right), \quad \text{s.t.} \quad \|\delta\|_p \leq \epsilon, \tag{2}$$

where $\delta^*$ is the best adversarial perturbation. $D(\cdot, \cdot)$ is the edit distance [19] between the two predicted texts, which is measured by counting the minimum number of character operations required to transform one string into the other.

In the black-box attack settings, the structure, parameters, and gradients of the victim model are not accessible. The attacker can query the victim model to get the model output, which could be either a probability distribution or a character sequence with confidence scores. In practice, most OCR systems provide only the latter, which contains less information for attackers. Considering practicality, we choose the latter attack scenario.

## 3.2 Objective decomposition

In pixel-level attacks, the final perturbation $\delta$ consists of a series of pixel perturbations $\delta_i$, *i.e.*, $\delta = \sum \delta_i$. And each pixel perturbation $\delta_i$ can be represented by a single attack pixel $p_i = (x_i, y_i, v_i)$, where $x_i$ and $y_i$ are the coordinates, and $v_i$ is the pixel value. Former methods [33, 37] optimize $\delta$ in

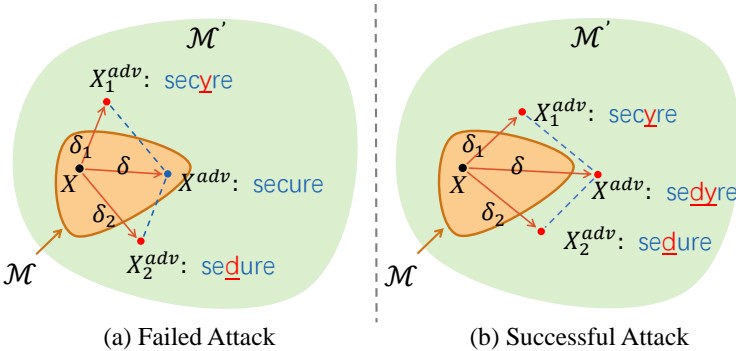

(a) Failed Attack            (b) Successful Attack

Figure 2: Illustration of the joint perturbation shift. (a) and (b) demonstrate two cases when combining two successful perturbations into one to attack text recognition models. Characters with underlines are incorrectly predicted.

the whole image space. This optimization paradigm is prone to fall into the local optimal solution where attack pixels are concentrated in the local space of the image. In this case, some characters will have no nearby attack pixel and then be correctly recognized.

To solve this problem, we decompose the optimization objective $\delta^*$ into multiple sub-objectives $\delta_i^*$, turning it to be a distributed optimization problem. Then we limit the perturbation location to a specific region in each sub-optimization problem. Specifically, we divide the image into multiple non-overlapping subspaces $\{\mathcal{R}_i\}_N$ and find a local optimal attack pixel $p_i$ in each subspace $\mathcal{R}_i$. In this way, a proper division of $\{\mathcal{R}_i\}_N$ could ensure that the attack pixels are not spatially concentrated.

### 3.3 The joint perturbation shift problem

In this distributed optimization problem, local optimization may undermine global feasibility. Even if each local optimized perturbation $\delta_i^*$ can successfully attack the target model $\mathcal{F}$, the global perturbation $\delta^* = \sum \delta_i^*$ has a certain probability of failing to attack, *i.e.*,

$$P\left(\mathcal{F}(X) = \mathcal{F}(X + \delta^*) | \bigwedge_{i=1}^N \mathcal{F}(X) \neq \mathcal{F}(X + \delta_i^*)\right) > 0. \tag{3}$$

We term this phenomenon the *joint perturbation shift* problem, and give a simple illustration of it in Figure 2. Let $\mathcal{M}$ and $\mathcal{M}'$ be the manifold area [1] where data samples can be correctly and incorrectly recognized by the target model, respectively. $X$ is the original data sample with label "*secure*". Although $X_1^{adv} = X + \delta_1$ and $X_2^{adv} = X + \delta_2$ are two successful adversarial samples located in $\mathcal{M}$, the joint perturbation $\delta = \delta_1 + \delta_2$ may shift the adversarial sample $X^{adv} = X + \delta$ to $\mathcal{M}'$ and lead to a failed attack.

## 4 Multi-population coevolution search

Following the distributed optimization paradigm, we propose the Multi-Population Coevolution Search (MPCS) method, which divides the image horizontally into $N$ equal regions $\{\mathcal{R}_i\}_N$ and utilizes evolutionary populations to search in each region. Meanwhile, a coevolution energy function is introduced to address the joint perturbation shift problem. Next, we will elaborate on the MPCS algorithm.

**Initialization.** We first randomly initialize an adversarial pixel population $\mathbf{P}_i$ in each region $\mathcal{R}_i$. Each population $\mathbf{P}_i$ contains $M$ adversarial pixels $\{p_i^j\}^M$, whose coordinates are randomly initialized and values are fixed to $\theta$, which can be formulated as follows,

$$\mathbf{P}_i = \{p_i^j = (x_i^j, y_i^j, v_i^j) | (x_i^j, y_i^j) \in \mathcal{R}_i, v_i^j = \theta\}^M, \tag{4}$$

where $x_i^j$ and $y_i^j$ are the pixel coordinates, $v_i^j$ is the pixel rgb value. Moreover, based on the finding from [37] that the adversarial pixel values of $\theta_{max} = (255, 255, 255)$ and $\theta_{min} = (0, 0, 0)$ have a

higher possibility to successfully attack than other values, we set $\theta$ as follows,

$$\theta = \begin{cases} \theta_{max}, & G(v_i^j) \leq \mu, \\ \theta_{min}, & G(v_i^j) > \mu, \end{cases} \tag{5}$$

where $G(v_i^j)$ denotes the gray value of $v_i^j$, and $\mu$ is set to 127 (the median of the set of pixel values). After that, we calculate the prior energy $\mathbf{E}^p$ of each adversarial pixel in populations as,

$$\mathbf{E}^p(p_i^j) = Dist(Y, \mathcal{F}(X + \Delta(p_i^j))), \tag{6}$$

where $\Delta(\cdot)$ is the perturbation generated by replacing the original pixel with the adversarial pixel, and $Dist(Y, \cdot)$ denotes the distance between the predicted label and the ground truth label $Y$ (detailed in Appendix D). Here, $\mathbf{E}^p$ is equal to the local objective function.

**Coevolution.** We let the $N$ populations coevolve in the direction of higher energy following the differential evolution (DE) algorithm [3, 37]. Briefly, each adversarial pixel $p$ in a population has a probability of mutating into a new adversarial pixel $p^{'}$. Among $p$ and $p^{'}$, the one with higher energy will be retained for the next generation.

However, as the existence of the joint perturbation shift problem, the prior energy function $\mathbf{E}^p$ cannot filter the best adversarial pixel. Therefore, we propose a novel coevolution energy function $\mathbf{E}^c$, which implicitly embeds the global constraint into the local objective function,

$$\mathbf{E}^c(p_i^j) = Dist(Y, \mathcal{F}(X + \Delta(\{p_i^j\} \cup \{p_k^*\}_{N-1})), \quad \text{s.t. } 1 \leq k \leq N, k \neq i, \tag{7}$$

where $\{p_k^*\}_{N-1}$ is the set of adversarial pixels with the highest energy in the other $N-1$ populations. Note that in the first generation, we select $\{p_k^*\}_{N-1}$ according to their prior energy $\mathbf{E}^p$. After the first generation, we replace $\mathbf{E}^p$ with $\mathbf{E}^c$ to calculate the energy of adversarial pixels. In this case, the evolution of one population will influence the other populations, which is similar to the coevolution phenomenon in nature.

**Early stop.** We propose an early stop mechanism, which terminates the evolution of the populations whose maximum energy has not increased compared to the previous generation. When the maximum number of generations $g_{max}$ is reached or when all populations have ceased evolution, we obtain the best adversarial pixels $\{p_i^*\}_N$ from each population.

**Post process.** If $\{p_i^*\}_N$ can successfully attack the victim model, we take them as the final attack pixels $\{p_i^{adv}\}_N$; otherwise, we will employ a pixel set population $\mathbf{P}_v$ to search for better pixel values following DE algorithm,

$$\mathbf{P}_v = \{\mathbf{Q}^t\}^L, \quad \mathbf{Q}^t = \{p_i^t|(x_i^t, y_i^t) = (x_i^*, y_i^*)\}_N, \tag{8}$$

where $L$ is the population size of $\mathbf{P}_v$, $\mathbf{Q}^t$ is the $t$-th adversarial pixel set whose pixel coordinates are frozen to be the same as $\{p_i^*\}_N$ and values are randomly initialized. The energy function of $\mathbf{P}_v$ is calculated as,

$$\mathbf{E}^v(\mathbf{Q}^t) = Dist(\mathcal{F}(X), \mathcal{F}(X + \Delta(\mathbf{Q}^t))). \tag{9}$$

At the max generation $g_{max}^{'}$ of $\mathbf{P}_v$, we obtain the best adversarial pixel set $\mathbf{Q}^{best}$ as the final attack pixels $\{p_i^{adv}\}_N$. With the final attack pixels, the adversarial example is generated as follows,

$$X^{adv} = X + \Delta(\{p_i^{adv}\}_N). \tag{10}$$

## 5   Analysis

We make a theoretical analysis of the superiority of our proposed coevolutionary energy function $\mathbf{E}^c$ over the prior energy function $\mathbf{E}^p$. For a certain pixel $p_i^j$ in population $\mathbf{P}_i$, we define:

$$A := \mathbb{I}\{\mathcal{F}(X) \neq \mathcal{F}(X + \Delta(p_i^j))\}, \quad \mathbf{E}^p(p_i^j) \propto P(A = 1), \tag{11}$$

where $\mathbb{I}$ is the indicator function whose value is 0 or 1, and the estimator $\mathbf{E}^p(p_i^j)$ approximately reflects the relative strength of the probability of $A = 1$. Similarly, we define:

$$B := \mathbb{I}\{\mathcal{F}(X) \neq \mathcal{F}(X + \Delta(\{p_i^j\} \cup \{p_k^*\}_{N-1})\}, \quad \mathbf{E}^c(p_i^j) \propto P(B = 1), \tag{12}$$

where $k \neq i$ and $\{p_k^*\}_{N-1}$ denotes the local optimal pixels in the other $N - 1$ populations.

Our final goal is to increase the expectation of $Z$, which is defined as,

$$Z := \mathbb{I}\{\mathcal{F}(X) \neq \mathcal{F}(X + \Delta(\{p_k^*\}_N))\}, \tag{13}$$

where $\{p_k^*\}_N$ denotes the local optimal pixels in all $N$ populations.

Next, according to the joint perturbation shift problem illustrated in Sec 3.3, we have:

$$P(Z = 1|A = 1) < 1, \quad P(Z = 0|A = 1) > 0, \tag{14}$$

which means the conditional distribution $P(Z|A = 1)$ is not a deterministic distribution. Therefore, the conditional entropy of $Z$ given $A = 1$ is strictly positive, *i.e.*,

$$H(Z|A = 1) = - \sum_{z \in \{0,1\}} P(Z = z|A = 1) log P(Z = z|A = 1) > 0. \tag{15}$$

On the other side, $\mathbf{E}^c(p_i^j) \leq \mathbf{E}^c(p_i^*)$ holds for all pixels in population $\mathbf{P}_i$. It means that if a successful attack can be achieved by the joint perturbation $\Delta(\{p_i^j\} \cup \{p_k^*\}_{N-1})$, then the joint perturbation of all the local optimal pixels $\Delta(\{p_k^*\}_N)$ must also yield a successful attack. Therefore:

$$P(Z = 1|B = 1) = 1, \quad P(Z = 0|B = 1) = 0. \tag{16}$$

Similarly, we have $H(Z|B = 1) = 0 < H(Z|A = 1)$, which reveals that when $B = 1$ (joint perturbation succeeds), the prediction of the global attack result $Z$ is completely deterministic, unlike when $A = 1$ (local perturbation succeeds). It means that the prior energy function $\mathbf{E}^p$ has a certain degree of information loss when estimating the attack performance of pixel $p_i^j$, while the coevolution energy function $\mathbf{E}^c$ fills the loss.

# 6 Experiments

## 6.1 Experiment settings

**Datasets and models.** We validate the effectiveness of our proposed attack method on four frequently-used datasets, including ICDAR13 [17] (horizontal regular scene text, 1015 images), SVTP [26] (perspective distorted text, 645 images), CUTE80 [28] (curved scene text, 288 images), and ICDAR15 [16] (irregular scene text, 2077 images). We fool three mainstream paradigms of scene text recognition, including the CTC-based paradigm (CRNN [29]), the attention-based paradigm (ASTER [30] and SAR [20]), and the multi-modal paradigm (IGTR [6]). All the models are from MMOCR [18] or OpenOCR [24]. If not specified, the experiments are conducted on the CRNN model and the CUTE80 dataset for convenience.

**Metrics.** We employ four metrics including the attack success rate before/after text correction (**SR/SR\***), the perturbation rate (**PR**), and the (**L2**) distance between the adversarial image and the original image. Specifically, we employ GPT-4o mini [14] to correct the mispredicted text results and then calculate the **SR\*** metric. The **PR** metric measures the proportion of the mispredicted characters,

$$\mathbf{PR} = \sum_{i=1}^{S} \frac{D(Y_i, \mathcal{F}(X_i^{adv}))}{len(Y_i)}, \tag{17}$$

where $S$ represents the total number of successful adversarial examples, $len(\cdot)$ denotes the length of the text, and $D(\cdot, \cdot)$ is the edit distance between the two predicted texts. The **L2** metric measures the image perturbation degree.

**Text correction.** Many STR methods use an extra dictionary to correct their prediction [22, 30, 13]. Language models can do the same job, and they are more flexible. So we choose to employ GPT-4o mini [14] for text correction and then calculate the **SR\*** metric. Further details are in Appendix C.

## 6.2 Comparison with pixel-level attacks

We replicate two state-of-the-art black-box pixel-level attack methods, $AD^2E$ [37] and OnePixel [33], which are also DE-based methods, and compare their attack performance with ours, as shown

Table 1: Comparison with other pixel-level attack methods. **SR\*** is the attack success rate after correction by GPT-4o mini [14]. **PR** denotes the character perturbation rate.

| Model | Method | CUTE80 | | | | IC13 | | | | SVTP | | | |
|---|---|---|---|---|---|---|---|---|---|---|---|---|---|
| | | SR. ↑ | SR*. ↑ | PR. ↑ | L2. ↓ | SR. ↑ | SR*. ↑ | PR. ↑ | L2. ↓ | SR. ↑ | SR*. ↑ | PR. ↑ | L2. ↓ |
| CRNN | OnePixel | 95.60 | 60.09 | 52.26 | 4.25 | 95.31 | 43.78 | 49.82 | 3.98 | 96.76 | 58.06 | 42.99 | 3.93 |
| | AD$^2$E | **95.60** | 49.73 | 29.75 | 4.20 | **96.31** | 36.24 | 29.77 | 3.97 | **96.99** | 42.61 | 23.50 | 3.91 |
| | MPCS | 87.36 | **86.75** | **94.54** | 4.30 | 85.47 | **70.12** | **90.19** | 4.02 | 85.65 | **77.67** | **80.07** | 4.04 |
| ASTER | OnePixel | **57.98** | 20.59 | 39.18 | 4.23 | 41.86 | 14.05 | 39.63 | 3.95 | **68.17** | 28.93 | 33.41 | 3.89 |
| | AD$^2$E | 57.14 | 16.93 | 28.67 | 4.22 | **47.66** | 13.48 | 32.94 | 3.96 | 67.45 | 22.35 | 22.06 | 3.92 |
| | MPCS | 51.26 | **27.77** | **74.93** | 4.32 | 43.31 | **21.60** | **83.05** | 3.98 | 57.17 | **42.14** | **51.79** | 3.95 |
| SAR | OnePixel | 65.98 | 19.26 | 43.40 | 4.20 | **42.52** | **16.16** | 41.40 | 3.91 | **58.45** | 18.89 | 28.67 | 3.87 |
| | AD$^2$E | **70.08** | 25.00 | 30.68 | 4.18 | 42.52 | 14.10 | 28.51 | 3.88 | 57.67 | 17.30 | 21.56 | 3.87 |
| | MPCS | 58.61 | **33.76** | **66.02** | 4.19 | 34.22 | 15.77 | **55.31** | 3.82 | 48.35 | **25.96** | **45.01** | 3.83 |
| IGTR | OnePixel | 41.38 | 24.29 | 61.55 | 4.35 | 24.27 | 17.19 | 55.66 | 4.02 | 41.41 | 24.04 | 54.22 | 4.03 |
| | AD$^2$E | 42.24 | 25.00 | 36.04 | **4.31** | 23.95 | 14.95 | 33.14 | **3.97** | 42.29 | 18.94 | 30.28 | **4.02** |
| | MPCS | **61.21** | **49.14** | **89.86** | 4.44 | **39.67** | **32.05** | **82.86** | 4.16 | **54.87** | **37.95** | **79.29** | 4.22 |

in Table 1. In specific, we set the number of attack pixels to 10 (the same as ours) when reproducing these methods. As shown in Table 1, our method achieves the highest perturbation rate (PR) on four mainstream STR models and three widely used datasets. For example, when attacking the SOTA STR model IGTR [6], our method achieves 89.86%, 82.86%, and 79.29% PRs on the CUTE80, IC13, and SVTP datasets respectively, surpassing the second-best method by 28.31%, 27.21%, and 25.07%.

Furthermore, when attacking the CRNN, ASTER, and SAR models on the CUTE80 dataset, our method achieves 26.66%, 7.18%, and 14.5% higher SR* metrics than the second-best method, although our SR metrics is slightly lower due to the joint perturbation shift. It demonstrates that our attack method is much more irreversible and that the PR metric can effectively evaluate the attack's irreversibility.

Table 2: Comparison with other attack paradigms. Sparse-rs [2] is a black-box attack method targeting image classification models. UDUP [4] is a transfer-based attack method targeting OCR systems. Xu et al. [36] propose a white-box attack method targeting STR models.

| Model | Method | CUTE80 | | | IC13 | | | SVTP | | |
|---|---|---|---|---|---|---|---|---|---|---|
| | | SR. ↑ | PR. ↑ | L2. ↓ | SR. ↑ | PR. ↑ | L2. ↓ | SR. ↑ | PR. ↑ | L2. ↓ |
| CRNN | Sparse-rs | 63.73 | 40.23 | 11.22 | 56.77 | 38.10 | 11.44 | 76.11 | 52.70 | 9.80 |
| | UDUP | 2.76 | 82.86 | **0.63** | 0.23 | 13.33 | **0.14** | 0 | 0 | **0.19** |
| | Xu et al. | 33.51 | 30.35 | 2.88 | 30.95 | 41.20 | 3.06 | 49.65 | 21.47 | 2.85 |
| | MPCS | **87.36** | **94.54** | 4.30 | **85.47** | **90.19** | 4.02 | **85.65** | **80.07** | 4.04 |
| ASTER | Sparse-rs | 12.55 | 7.65 | 11.41 | 9.50 | 6.84 | 11.46 | 22.18 | 11.27 | 9.88 |
| | UDUP | 0.97 | 30.00 | **1.20** | 0.12 | 16.67 | **0.15** | 0 | 0 | **0.20** |
| | Xu et al. | 36.26 | 35.38 | 2.85 | 32.84 | 38.70 | 3.04 | 56.29 | 23.04 | 2.82 |
| | MPCS | **51.26** | **74.93** | 4.32 | **43.31** | **83.05** | 3.98 | **57.17** | **51.79** | 3.95 |
| SAR | Sparse-rs | 19.23 | 11.65 | 11.43 | 12.47 | 9.54 | 11.45 | 24.77 | 12.03 | 9.76 |
| | UDUP | 3.70 | **107.45** | **1.78** | 0 | 0 | **0.20** | 0.18 | 16.67 | **0.17** |
| | MPCS | **58.61** | 66.02 | 4.19 | **34.22** | **55.31** | 3.82 | **48.35** | **45.01** | 3.83 |

## 6.3 Comparison with other attack paradigms

To demonstrate the superiority of our proposed method in attacking STR models, we compare our method with several other attack paradigms, as shown in Table 2. Xu et al. [36] propose a white-box attack method that updates the perturbation iteratively by optimization. When reproducing its code, we set the upper limit of the L2 metric to 10 to prevent image distortion caused by excessive perturbation, while our L2 metrics are less than 5. On the CRNN model and CUTE80 dataset, the SR and PR of this method are 53.85% and 64.19% lower than ours. This is because this method can hardly find successful perturbations under low perturbation limitations.

Table 3: Ablation study of our method. **MP** means the multi-population mechanism, **CEF** denotes the coevolution energy function, **MG** represents that the population will evolve to the maximum generation (*i.e.*, it will not stop when the attack is successful), and **ES** is the early stop mechanism.

| Method | MP | CEF | MG | ES | SR. ↑ | SR*. ↑ | PR. ↑ | L2. ↓ |
|---|---|---|---|---|---|---|---|---|
| Baseline | | | | | **95.6** | 55.58 | 29.75 | **4.20** |
| MPCS | ✓ | | | | 76.37 | 52.95 | 52.37 | 4.32 |
| | ✓ | ✓ | | | 88.46 | 75.42 | 71.43 | 4.30 |
| | ✓ | ✓ | ✓ | | **90.66** | **87.20** | **99.72** | 4.29 |
| | ✓ | ✓ | ✓ | ✓ | 87.36 | 86.75 | 94.54 | 4.30 |

Besides, we compare our method with UDUP [4], a transfer-based attack method that fools OCR models by generating adaptive underpainting. As shown in Table 2, this method has extremely low SR when attacking STR models, up to 84.6% lower than ours. This is because UDUP cannot successfully add underpainting to scene text images with complicated backgrounds. Furthermore, we adopt Sparse-rs [2], a black-box attack method designed for image classification models, for attacking text recognition models by modifying its victim models and loss function. Our method outperforms Sparse-rs with +23.63% SR, +54.31% PR, and -6.92 L2 on CUTE80 and CRNN. These results demonstrate that attack methods tailored for image classification models are not suitable for attacking text recognition models.

### 6.4 Ablation study

In this section, we conduct comprehensive ablation experiments on the four main components of our proposed method. We reproduce $AD^2E$ [37] under our experiment settings and use it as our baseline. As shown in Table 3, after replacing the single population in the baseline with our proposed multi-population (**MP**), the PR increases by 22.62% while the SR decreases by 19.23% (due to the joint perturbation shift problem). As shown in Figure 3, with the spatial constraints imposed by **MP**, adversarial pixels within each population gradually converge to small regions that are evenly distributed across the entire image during the evolution process. Next, we employ the coevolution energy function (**CEF**) to get SR and PR increased by 12.09% and 19.06%, demonstrating its effectiveness.

Furthermore, in order to achieve a better performance, we force all the populations to evolve until the maximum generation (**MG**) rather than stopping evolution when the attack is successful. The fourth row shows that this solution gains an improvement of 2.2% and 28.29% on SR and PR, respectively. Meanwhile, we propose the early stop (**ES**) mechanism to balance the extra time consumption caused by **MG**. According to the last row in Table 3, it has a small impact on the attack performance. And the average total number of evolutionary generations with ES is less than that without ES, as shown in Figure 4.

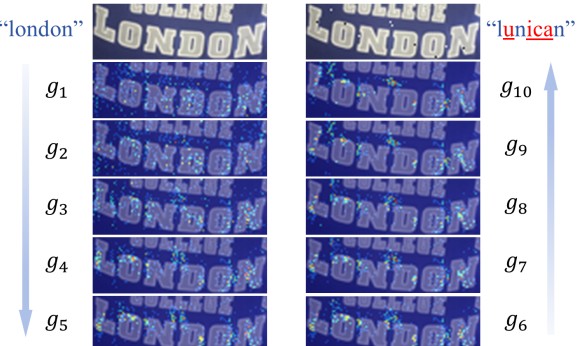

Figure 3: Population evolution visualization.

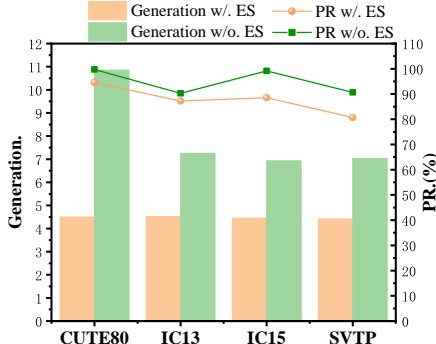

Figure 4: Effect of the early stop (ES).

Table 4: Influence of the population number $N$ and the population size $M$. We set up 6 experiments on two datasets while keeping the sum of all population sizes constant.

| $N, M$ | CUTE80 | | | IC13 | | |
|---|---|---|---|---|---|---|
| | SR.↑ | PR.↑ | L2.↓ | SR.↑ | PR.↑ | L2.↓ |
| (4, 150) | 75.27 | 61.84 | **2.75** | 69.60 | 56.19 | **2.59** |
| (5, 120) | 77.47 | 70.23 | 3.09 | 73.19 | 60.75 | 2.90 |
| (6, 100) | 84.61 | 71.83 | 3.37 | 74.41 | 63.93 | 3.17 |
| (10, 60) | 87.35 | 94.54 | 4.30 | 80.11 | 87.23 | 4.02 |
| (20, 30) | **91.20** | **124.52** | 6.03 | **88.27** | **133.77** | 5.72 |

## 6.5 Influence of hyper-parameters

Firstly, we investigate the influence of the population number $N$ and the population size $M$ of $\mathbf{P}_i$, as shown in Table 4. We keep the sum of all population sizes constant, *i.e.*, $N \times M = 600$, and adjust the combination of $N$ and $M$. As $N$ increases to 20 on CUTE80, the SR, PR, and L2 rise to 91.20%, 124.52%, and 6.03, respectively. We adopt $N = 10$, which also has a promising attack performance with smaller perturbation.

## 6.6 Correction by the dataset dictionary

We also conduct experiments on a more traditional text correction method, *i.e.*, through an extra dictionary. We use the test set vocabulary of the IC15 dataset as the dictionary to correct the results of successful attacks through it. As shown in Table 5, among the attack results on the three STR models, our method exhibits much higher SR and PR compared to both OnePixel [33] and $\mathrm{AD^2E}$ [37]. These results demonstrate that our method still has better attack irreversibility under this setting.

## 6.7 Effect range of adversarial pixels

We conduct experiments to investigate the effect range of adversarial pixels. In specific, we restrict the location of adversarial pixels to ensure that some characters do not have adversarial pixels around them. As shown in Table 6, when the adversarial pixels are restricted to the left half and right half of the image, the PR metric decreases by 20.42% and 13.68%, respectively. These results demonstrate that the adversarial pixel has a higher possibility of affecting nearby characters rather than farther characters.

Table 5: The attack results on the IC15 dataset. **SR\*** is the attack success rate after correction by the dataset dictionary.

| Method | CRNN | | ASTER | | SAR | |
|---|---|---|---|---|---|---|
| | SR*.↑ | PR.↑ | SR*.↑ | PR.↑ | SR*.↑ | PR.↑ |
| OnePixel | 42.15 | 77.93 | 19.19 | 62.18 | 21.10 | 65.20 |
| $\mathrm{AD^2E}$ | 29.63 | 50.31 | 18.32 | 51.17 | 17.23 | 50.24 |
| MPCS | **71.19** | **94.48** | **35.53** | **78.27** | **27.84** | **73.87** |

Table 6: The attack performance on the CUTE80 dataset when the perturbation area is limited to the left and right half of the image, respectively.

| Perturbation Area | SR.↑ | PR.↑ | L2.↓ |
|---|---|---|---|
| Left | 32.97 | 74.12 | 3.38 |
| Right | 39.01 | 80.68 | 3.65 |
| All | **87.36** | **94.54** | **4.30** |

## 7 Discussion and limitations

In this work, we propose a pixel-level black-box attack method targeting at altering each character in STR models' prediction, termed as MPCS. By modifying only a few pixels in the image, MPCS can mislead STR models to predict more incorrect characters than previous methods, while maintaining the whole visual semantic information. We believe MPCS offers valuable applications while requiring careful consideration of its societal implications. On the positive side, the generated adversarial examples can strengthen STR models through adversarial training, improving their robustness against real-world perturbations. Additionally, MPCS enables effective privacy protection by preventing

unauthorized OCR extraction of sensitive text with minimal visual distortion. For example, when users upload images to social platforms, we could add tiny perturbation pixels to the images so that the private texts may not be correctly recognized and collected by malicious OCR systems. Besides, the potential misuse of such adversarial attacks raises important concerns. Malicious actors could exploit MPCS to bypass security systems, manipulate automated text recognition, or evade content moderation, highlighting the need for responsible research practices and countermeasures.

Meanwhile, although MPCS does not affect the viewing of text images by human eyes, the pixels it disturbs are still quite different from the original pixels. In future work, we will investigate finding a better pixel value optimization method to minimize the deviation between adversarial and original pixel values, thus enhancing the attack imperceptibility.

## Acknowledgments and Disclosure of Funding

This work was supported by the National Natural Science Foundation of China (62302532, 62411540034), the Natural Science Foundation of Guangdong (2025A1515011224), and Shenzhen Science and Technology Program (202206193000001, 20220816225523001, KQTD20221101093559018).

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

# Appendix

## A    Implementation details

In our experiments, we set the population number $N$ and the population size $M$ to 10 and 60, respectively. For population $\mathbf{P}_v$, we set the population size $L$ to 600. The max generation $g_{max}$ and $g'_{max}$ are set to be 10 and 1 for a low query budget. The height $h$ and width $w$ of the input image are 32 and 100, respectively. The experiments are conducted on a workstation with a single GeForce RTX 3090 GPU, a 2.90 GHz Intel(R) Xeon(R) Gold 6226R CPU, and 64G RAM, based on Pytorch v2.0.1.

## B    Visualization

We visualize our adversarial examples and their corresponding attack results, and compare them with other attacks, as shown in 5. Furthermore, we visualize the generated adversarial examples against the CRNN model under different settings in Table 4, as shown in Figure 6. Typically, the attack performance becomes better with the increasing $N$. Besides, we notice that only a few pixels are needed to attack a single character, as shown in the second and third rows. Therefore, we can dynamically adjust the attack pixel number according to specific images to weaken the perturbation in practice.

| Method | Origin | Xu et al. | OnePixel | AD$^2$E | Ours |
|---|---|---|---|---|---|
| Image | | | | | |
| Label | "carp" | "calp" | "cairle" | "calp" | "doeilits" |
| Image | | | | | |
| Label | "islands" | "islan_s" | "iseaneys" | "isbands" | "sseanidng" |
| Image | | | | | |
| Label | "estd" | "esti" | "esitip" | "estp" | "fisitrio" |
| Image | | | | | |
| Label | "tokyo" | "tokya" | "tokyje" | "tokye" | "iokije" |
| Image | | | | | |
| Label | "20" | "200" | "saoy" | "200" | "geallan" |
| Image | | | | | |
| Label | "dreams" | "dreans" | "drenims" | "dreans" | "brering" |

Figure 5: Visualization of the adversarial examples and their attack results. Characters with underlines are incorrectly predicted. Compared to other methods, our approach exhibits extraordinary attack performance. Zoom in to get a better view.

## C    Text correction by LLM

We employ GPT-4o mini [14] to correct the mispredicted text results and calculate the SR* metric. The prompt is formed as follows:

"Correct the spelling of the following words:

slacrest =>

| N | 4 | 5 | 6 | 10 | 20 | Origin |
|---|---|---|---|---|---|---|
| Image | CLUB | CLUB | CLUB | CLUB | CLUB | CLUB |
| Label | "callb" | "callb" | "aalld" | "labls" | "fabties" | "club" |
| Image | 14 | 14 | 14 | 14 | 14 | 14 |
| Label | "a444" | "hally" | "nidely" | "maaf" | "alail" | "14" |
| Image | 6 | 6 | 6 | 6 | 6 | 6 |
| Label | "cess" | "cod" | "cod" | "cess" | "geswi" | "6" |
| Image | PUBLICK | PUBLICK | PUBLICK | PUBLICK | PUBLICK | PUBLICK |
| Label | "rubblack" | "plislnek" | "rubbbick" | "fuistrer" | "foishaotes" | "publick" |

Figure 6: Visualization of our adversarial examples against the CRNN model under different attack settings. $N$ is the population number, which is equal to the attack pixel number. The sum of population sizes is 600 in all experiments.

peach =>

staron =>

......"

The output of the model may be like:

"Here's the corrected list of words:

slacrest => Seacrest

peach => peach (unchanged)

staron => station

......"

After we get the model output, we first remove the annotations like "(unchanged)" to avoid their influence. Then we convert all words to lowercase and compare the predicted words with corresponding labels in lowercase. If the predicted word is different from its label, we treat it as a failed attack; otherwise, it is a successful attack.

## D   The calculation of distance

We employ CTC Loss [10] to calculate the distance $Dist(Y, \mathcal{F}(X^{adv}))$. $Y = [y_1, y_2, ..., y_n]$ ($n$ means the gt character number) is the ground truth label, and $\mathcal{F}(X^{adv})$ denotes the model output. Since we employ the black-box attack setting where the model output $\mathcal{F}(X^{adv})$ includes a character sequence $[\hat{y}_1, \hat{y}_2, ..., \hat{y}_m]$ ($m$ means the predicted character number) and a confidence score sequence $[\hat{s}_1, \hat{s}_2, ..., \hat{s}_m]$ ($s \in [0, 1]$), we need to preprocess it to facilitate the CTC loss calculation.

Firstly, we define the character dictionary $Dict$. Next, we generate a $m \times (l + 1)$ matrix $T$ as,

$$T[a, b] = \begin{cases} \hat{s}_a, & Dict(\hat{y}_a) = b, \\ \dfrac{1}{l+1}, & Dict(\hat{y}_a) \neq b, \end{cases} \tag{18}$$

where $Dict(\hat{y}_a)$ represents the index of character $\hat{y}_a$ in the dictionary. Then we apply the log-softmax function along the second dimension of tensor $T$ to obtain the matrix $T^*$ of logarithmic probabilities. After that, we can calculate the CTC loss between $T^*$ and $Y$.

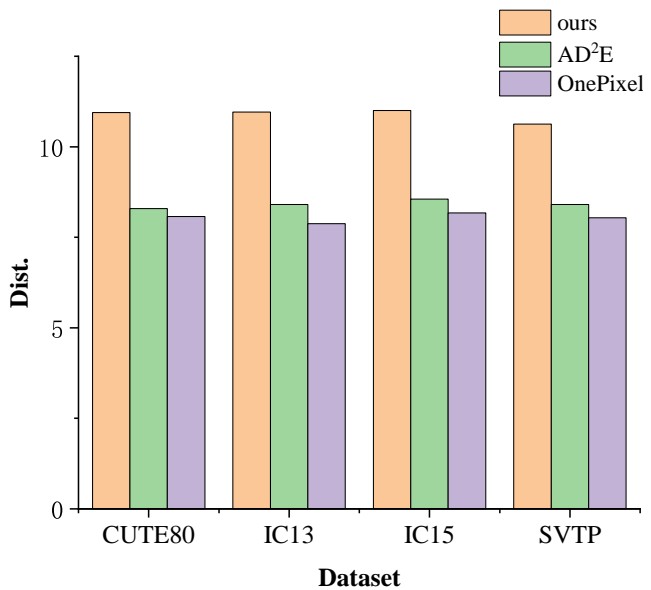

Figure 7: The average shortest distance between adversarial pixels on the CRNN model.

| | Method | Origin | OnePixel | AD$^2$E | MPCS |
|---|---|---|---|---|---|
| **CRNN** | **Image** | | | | |
| | **Label** | "disneyland" | "disneyla_i_nd" | "disneyl_e_nd" | "_orsuagiera_" |
| **ASTER** | **Image** | | | | |
| | **Label** | "disneyland" | "dist_ee_land" | "disney_f_and" | "_otsk_eyi_and_" |
| **SAR** | **Image** | | | | |
| | **Label** | "disneyland" | "dis_:i_eyland" | "disney_f_and" | "_01.5.2_yl_2_nd_" |

| | Method | Origin | OnePixel | AD$^2$E | MPCS |
|---|---|---|---|---|---|
| **CRNN** | **Image** | | | | |
| | **Label** | "chocolates" | "choi_col_ol_es" | "chocol_oj_es" | "_shiacidloied_" |
| **ASTER** | **Image** | | | | |
| | **Label** | "chocolates" | "choc_a_lates_._" | "cha_ce_lates" | "chi_aca_iates" |
| **SAR** | **Image** | | | | |
| | **Label** | "chocolates" | "ch_a_colates" | "choc_c_lates" | "_thetc_iates" |

Figure 8: Visualized comparison on the CRNN, ASTER, and SAR models. The underlined characters are incorrectly predicted. The red rectangles in images outline the characters with no nearby adversarial pixel. Zoom in to get a better view.

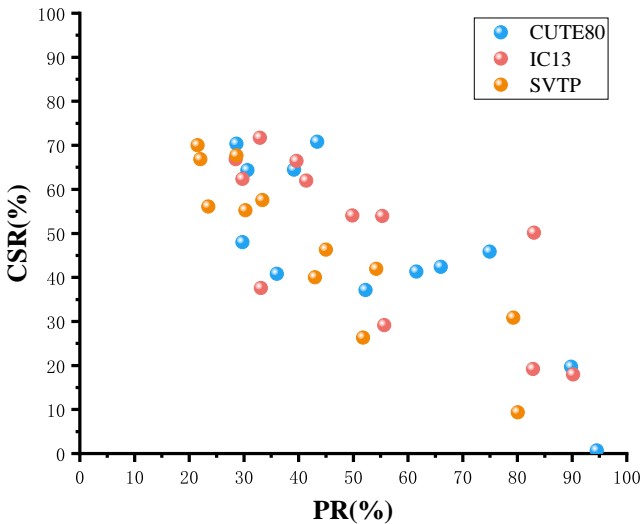

Figure 9: The relation between the correction success rate (CSR) by GPT-4o mini and the perturbation rate (PR) of texts.

## E  Attack pixel spatial concentration problem

In this section, we demonstrate the attack pixel spatial concentration problem that exists in previous pixel-level attack methods [33, 37]. Firstly, we give a quantitative analysis by calculating the average shortest distance between adversarial pixels. Given $N$ adversarial pixels $[p_1, p_2, ..., p_N]$, we calculate the distance between each adversarial pixel and its nearest adversarial pixel to get $[d_1, d_2, ..., d_N]$, and then calculate the average value of $d$. As shown in Figure 7, our method has greater average distances on four datasets than previous methods. This is because our proposed multi-population mechanism restricts the adversarial pixel location in the horizontal direction.

Furthermore, we give a qualitative comparison by visualizing the attack results of OnePixel [33] and $AD^2E$ [37] with ours on the CRNN [29], ASTER [30], and SAR [20] models, respectively. The attack pixel number is 10. As shown in Figure 8, both OnePixel and $AD^2E$ have the adversarial pixel local aggregation problem when attacking the three STR models. The attack pixels concentrate within localized regions of the image, resulting in some characters having no attack pixels nearby. Consequently, these characters can be correctly recognized by the model. Meanwhile, they can only alter one or two predicted characters in most cases. Differently, our method effectively solves the adversarial pixel local aggregation problem through the multi-population mechanism. The attack pixels we generated are evenly distributed across the entire image along the horizontal direction, ensuring that each character in the image has at least one nearby attack pixel.

## F  The perturbation rate metric

To evaluate whether PR metrics can accurately reflect the irreversibility of attacks, we visualize the relationship between the correction success rate (CSR) by GPT-4o mini and the perturbation rate (PR) of texts. As shown in Figure 9, on the CUTE80 [28], IC13 [17] and SVTP [26] datasets, the overall correction success rate decreases as the perturbation increases. It demonstrates the effectiveness of the perturbation rate.

## G  Transfer attack on commercial OCR system

We implement transfer attacks on BaiduOCR (`https://cloud.baidu.com/doc/OCR/index.html`) using adversarial examples that can successfully attack SAR. As shown in Figure 10, the transfer attack achieves 53.44%, 63.51%, and 62.65% LPR on IC13, IC15, and SVTP datasets, respectively. Some qualitative results are also shown in Figure 10. These results demonstrate that our

| Dataset | IC13 | IC15 | SVTP |
|---------|------|------|------|
| PR | 53.44 | 63.51 | 62.65 |
| Image |  |  |  |
| Pred | for → a | premier → b | kyro → u |
| Image |  |  |  |
| Pred | 30 → e | soymilk → 50 | beads → beal |

Figure 10: The result of conducting a transfer attack on BaiduOCR using adversarial samples obtained by attacking the SAR model.

$X$:     $Y$: "*seacrest*"

$X^{adv}$:     $F(X^{adv})$: "*slagrese*"

Figure 11: Metric computation example.

method has good transferability and is still capable of misleading the model to predict a significant number of incorrect characters during the transfer attack.

## H   Metric computation example

We provide a real metric computation example in this section. As shown in Figure 11, the original image $X$ has a predicted label $Y$ = "*seacreast*", and the adversarial example image $X^{adv}$ is recognized by the STR model $\mathcal{F}$ as $\mathcal{F}(X^{adv})$ = "*slagrese*". Then, the edit distance between their predicted labels is $D(Y, \mathcal{F}(X^{adv})) = 3$, and the length of the ground truth is $len(Y) = 9$. Therefore, the perturbation rate (PR) of this adversarial example is $PR = \frac{D(Y, \mathcal{F}(X^{adv}))}{len(Y)} = 33.33\%$. Besides, the L2 metric evaluates the average L2 distance between the original image and the adversarial example image: $\mathbf{L2} = \frac{1}{S} \sum_{i=1}^{S} ||X_i - X_i^{adv}||_2$. In this example, the L2 distance is 3.62. As for the success rate before/after correction (SR/SR*), for example, if the number of samples is 1000, and 800 of them are successfully attacked, then the SR is 80%. Next, we use an LLM to correct the mispredicted texts. If 200 of them cannot be corrected, then the SR* is 20%.

## I   Evaluation on non-Latin scripts

To further examine the generality of our proposed method beyond Latin scripts, we additionally conducted experiments on a Chinese text recognition benchmark. Our main experiments rely on STR models provided by MMOCR and OpenOCR, both of which are trained on large-scale English datasets. Consequently, our primary evaluation focuses on four widely used English benchmarks.

However, as our method does not depend on any script-specific assumptions, we expect it to generalize to non-Latin scripts as well. To validate this, we performed a comparative study on the dataset introduced in "Benchmarking Chinese Text Recognition: Datasets, Baselines, and an Empirical Study." Specifically, we randomly sampled 200 images from this benchmark and applied both our attack and the baseline method to the IGTR model, which was trained on Chinese datasets by OpenOCR. Our method achieved a success rate (SR) of 51.85% and a perturbation rate (PR) of 34.44%, outperforming the baseline (SR: 38.89%, PR: 21.12%). These results demonstrate that our approach maintains strong effectiveness on non-Latin scripts, further confirming its generalizability across different languages and writing systems.

