# OpenReview forum: "Towards Irreversible Attack: Fooling Scene Text Recognition via Multi-Population Coevolution Search"
_NeurIPS.cc/2025/Conference — NeurIPS 2025 poster_

### Official Review · Reviewer_HCsA · 2025-06-30

**Clarity:** 3
**Significance:** 2
**Originality:** 2
**Rating:** 3
**Confidence:** 3

**Summary:**

Against adversarial attacks on Scene Text Recognition (STR) models, existing attack methods generate erroneous text that is easily corrected by a dictionary or a Large Language Model (LLM), i.e., the attack is “reversible”. To address this problem, the authors aim to generate “irreversible” attacks, that is, to allow the model to predict as many incorrect characters as possible. Therefore, this paper proposes a black-box pixel-level attack method called “Multi-Population Co-Evolutionary Search” (MPCS). The authors first solve the problem of “spatial concentration of attack pixels” in previous attacks by decomposing the global optimization objective into multiple image subspaces. Then, in order to solve the emerging problem of “combined perturbation shift” due to the objective decomposition, the authors design an innovative “co-evolutionary energy function”, which enables the parallel pixel populations in different subspaces to co-evolve. Finally, the experimental results show that MPCS can significantly increase the number of mispredicted characters, thus realizing more destructive and irreversible attacks.

**Questions:**

1.	Is “θmin = (0, 0, 0) and θmin = (0, 0, 0)” a wrong representation in the initialization of pixel values section? In addition, the authors skip the description of this paragraph only by citing the literature [38]. The unreasonable representation in both aspects makes it more difficult for readers to understand the methodology of this paper.

2.	I believe that the essence of the attack task is to find a trade-off between the conflicting goals of “maximizing the success rate of the attack” and “minimizing the image perturbation”. Therefore, why don't the authors choose a multi-objective optimization framework to obtain a series of Pareto-optimal solutions covering different preferences?

3.	For a black-box attack algorithm, the computational cost is also crucial. However, the authors propose a complex framework containing multi-population, multi-generation evolution, whose computational cost is obviously huge. Can the authors provide relevant explanations and experimental analysis?

4.	The coevolution energy function (CEF) relies on a single optimal individual for each other population. Is this robust enough for the evolutionary process? Have the authors considered other coevolution strategies, such as using Top-K individuals from each population to represent its evolutionary state?

5.	Can the authors provide more explanation on “why optimization-based white-box methods are not effective under low L2 paradigm constraints” when MPCS outperforms constrained white-box attack methods in their experiments?

**Ethical Concerns:**

["NO or VERY MINOR ethics concerns only"]

**Limitations:**

Yes

**Quality:**

2

**Strengths And Weaknesses:**

Strengths:

The authors do not merely allow the model to produce incorrect predictions, but also introduce the concept of “irreversibility” of the attack. This provides a good point for the study of adversarial attacks against sequence prediction models.

Weakness:

The contribution of this paper lies in solving two core problems: “pixel concentration” and “combined perturbation shift”. The former is just a well-known local optimization problem in evolutionary algorithms. The latter, i.e., the “Combined Perturbation Shift Problem”, which is emphasized by the authors, is extremely abstract and unclear. Although a definition is given in the paper, the concept is still not well understood by the reader according to the short description in Section 3.3 and Figure 2.

The high computational cost of the MPCS method is obvious, especially when evaluating the coevolutionary energy function, where each individual needs to be evaluated in combination with the optimal individuals of the other N-1 populations. The authors do propose an early stop mechanism to balance performance and efficiency. However, this is only a mitigation measure and does not address the fundamental problem of high computational costs associated with evolutionary algorithms.

The field of black-box attacks is rapidly developing, and it is possible that other more powerful or varied SOTA algorithms exist. However, it is hardly convincing that the authors chose only OnePixel and AD²E, two obsolete methods also based on evolutionary algorithms, as comparative methods for pixel-level attacks.

---

> ### Author Rebuttal · Authors · 2025-07-31
>
> **W1: Combined perturbation shift problem.**
> We thank the reviewer for the valuable feedback.
> The combined perturbation shift refers to the phenomenon: when we combine several successful perturbations $ \delta_i^* $  to get a new perturbation $ \delta^* = \sum \delta_i^* $, this new perturbation could possibly be a failed perturbation.
> This is the key challenge we want to solve.
> Due to space constraints, our explanation in Section 3.3 and Figure 2 was brief; we will provide a more intuitive illustration and detailed discussion in the camera-ready version to make the concept clearer.
>
> **W2/Q3: The high computational cost.**
> We thank the reviewer for the valuable feedback.
> The computational cost mainly comes from querying the target STR model, while the DE algorithm itself only needs little computation.
> Also, when evaluating the coevolutionary energy function $\textbf{E}^c(p_i^j)$, we only query the model once using the combined perturbation $\Delta(\lbrace p_i^j \rbrace \cup \lbrace p_{k}^* \rbrace _{N-1})$, which has the same query cost as $\textbf{E}^p(p_i^j)$.
> Thus, the coevolutionary energy function does not introduce any external computational cost.
>
> Besides, we set the total attack pixel number consistent across different multi-population settings.
> For example, when the population number $N=10$, the population size is $M=60$; when $N=1$, the population size is $M=600$.
> Therefore, no matter how many populations we used, the computational cost in one generation is consistent.
> In our setting, the average attack time on CRNN with CUTE80 is only 18.23 seconds, which we consider acceptable.
>
> **W3: Comparative methods.**
> We thank the reviewer for the constructive feedback. The field of black-box attacks indeed evolves rapidly.
> We chose OnePixel and $\mathrm{AD}^2\mathrm{E}$ because they are well-established baselines for pixel-level black-box attacks, sharing the same evolutionary paradigm as our method.
> To the best of our knowledge, $\mathrm{AD}^2\mathrm{E}$ remains the state-of-the-art among pixel-level black-box attacks in the STR domain.
> In addition, we included recent SOTA black-box attacks such as Sparse-RS and UDUP in Table 2, although these methods are either not pixel-level or not directly tailored to STR tasks.
> Therefore, we believe our comparison covers both representative baselines within the same paradigm and other strong black-box attack strategies across paradigms.
>
>
> **Q1: Pixel value initialization.**
> We thank the reviewer for highlighting this error.
> The " $\theta_{min}=(0,0,0)$ and $\theta_{min}=(0,0,0)$" should be "$\theta_{min}=(0,0,0)$ and $\theta_{max}=(255,255,255)$".
> We chose these two values because they are found to be most effective in the work [38].
>
> **Q2: Optimization framework.**
> We thank the reviewer for this inspiring question.
> Indeed, the attack task consists of two objectives: maximizing the success rate and minimizing the image perturbation.
> Therefore, in future work, we can use a multi-objective optimization framework, such as NSGA-II, to find the best attack pixel values.
> Specifically, after we obtain the attack pixel locations via MPCS, we can use NSGA-II to search for attack pixel values that are most imperceptible.
> **This approach complements rather than conflicts with our MPCS method**, and we plan to explore it as a promising direction in future work.
>
> **Q4: Coevolution strategy.**
> We thank the reviewer for this insightful question.
> As shown in Table 2, when introducing the CEF, the SR, SR*, and PR get a rise of 12.09\%, 22.47\%, and 19.06\%, respectively.
> This demonstrates its effectiveness and robustness.
> As for the coevolution strategy mentioned by the reviewer, we believe it will enhance the attack performance with several times the amount of calculation.
> With CEF, we calculate the energy of an individual by generating a combined perturbation $\Delta(\lbrace p_i^j \rbrace \cup \lbrace p_{k}^* \rbrace_{N-1})$ and use it to query the target model one time.
> If we use the top-K individuals from other populations, there are two choices:
>
> (1) Using all top-K individuals to generate a combined perturbation $\Delta(\lbrace p_i^j \rbrace \cup \lbrace \lbrace p_{k}^s \rbrace^K \rbrace _{N-1})$.
> This will increase the perturbation, making it not aligned with our final perturbation.
> The increased perturbation will also bring a much higher SR.
> So it is not a good choice.
>
> (2) Traversing the top-K individuals to generate combined perturbations one by one.
> This may enhance the robustness, but since each combined perturbation needs a query time, the computational cost will increase several times.
>
> **Q5: Explanation on white-box methods.**
> We thank the reviewer for this good question.
> We reproduced the white-box method [37] and found that if we set no upper bound on its perturbation size, some of the adversarial examples generated by it could be severely distorted, but the average perturbation is still small.
> In contrast, our method has a stable perturbation size for each adversarial example.
> For a fair comparison, we have to set an upper bound on the perturbation size of [37].

---

### Official Review · Reviewer_1ydv · 2025-07-03

**Clarity:** 3
**Significance:** 2
**Originality:** 3
**Rating:** 4
**Confidence:** 4

**Summary:**

This article proposes a novel black box adversarial attack method - MPCS, for STR models. There are two key issues with existing adversarial attack methods: Information residue, the predicted results after the attack still retain some of the original text information, which is easily corrected by language models; Pixel concentration leads to uneven spatial distribution of perturbed pixels, resulting in some characters not being effectively attacked. The core contribution of this article is to achieve a globally optimized distribution of perturbed pixels through horizontal partitioning and co evolutionary energy functions; Propose and solve the combined perturbation offset problem and prove the information lossless nature of the energy function.

**Questions:**

I am curious about how formula 7 balances local optimization and global optimization? Will this process fall into the dilemma of local optimality?

Table 4 shows that when N=20, L2=6.03. Will non-uniform partitioning reduce distortion?

In fact, the disturbance points generated by this method are obvious, although the overall disturbance value is not high. Is there a way to make the disturbance points more hidden?

Can the number of populations and the size of populations be automatically adjusted? This is more conducive to the optimization of this method.

**Ethical Concerns:**

["NO or VERY MINOR ethics concerns only"]

**Final Justification:**

The responses have mostly addressed my concerns. Based on the feedback from the authors and the other reviewers' comments, I would like to keep my original score as BA.

**Limitations:**

Yes.

**Quality:**

3

**Strengths And Weaknesses:**

The paper presents a systematic attack framework MPCS that tackles a critical challenge in adversarial attacks on STR models: irreversibility. The multi-population coevolution search is well-motivated and addresses the pixel concentration problem effectively.

## Strengths :

The authors tested it on 4 standard datasets (ICDAR13/15, SVTP, CUTE80) and 3 STR model types (CTC-based, attention-based, multimodal). At the same time, it was evaluated against 6 baseline models (OnePixel, AD²E, Sparse-rs, UDUP, Xu et al., transfer attack). The perturbation rate (PR) was introduced to measure the irreversibility of the attack and verified by correction based on GPT-4.

This paper explicitly models irreversibility in STR attacks, going beyond simple misclassification. The authors formally define the combined perturbation shift problem and prove that the co-evolutionary energy function can avoid information loss. Meanwhile, MPCS ensures that the perturbation is evenly distributed. The early stopping mechanism balances effectiveness and efficiency.

The authors demonstrate that even state-of-the-art STR models (e.g., IGTR) are vulnerable to irreversible attacks and propose how adversarial examples can protect sensitive text (e.g., license plates, ID cards) from unauthorized OCR extraction.

## Weaknesses：

The performance of the method proposed by the author is reduced for long texts. It is only tested on the CNN/RNN-based STR model, and no experiments are done on Vision Transformers or diffusion-based OCR models to verify the effect of the method. Besides, lack of exploration on the defense mechanism of this method.

Formula 7 does not clearly describe the connection between local optimization and global optimization. Can it be theoretically guaranteed that the method finds the global optimum without falling into the local optimum?

At the same time, this article does not consider whether the current mainstream defense measures can defend against the method of this model. For example, can the adaptive STR model detect MPCS perturbations?

Although this article emphasizes privacy protection, the scenario of malicious applications can be better discussed.

What are the usage scenarios of this method and is there any possibility of its application in real-life scenarios?

---

> ### Author Rebuttal · Authors · 2025-07-31
>
> **W1: Reduced performance on long texts.**
> We thank the reviewer for highlighting this point.
> We conduct some experiments on CRNN and CUTE80 to explore the performance on long texts and short texts under the same setting.
> When the text length is greater than 4, the success rate (SR) is 84.96\% and the perturbation rate (PR) is 63.33\%.
> In contrast, for texts with length $\leq 4$, the SR is 89.86\% and PR is 134.29\%.
> This indicates that longer texts are indeed harder to attack, as the number of attack pixels per character decreases with text length.
> However, as shown in Table 4, increasing the total number of attack pixels can effectively improve performance on long texts.
>
> **W2: Attack on transformer or diffusion-based models.**
> We thank the reviewer for this question.
> As shown in Table 1, we have already conducted experiments on IGTR (TPAMI 2025), which is a multi-modal STR model with a transformer architecture. As for the diffusion-based STR model, to the best of our knowledge, there is no open-sourced diffusion-based STR model to implement experiments on.
>
> **W3: Against defense mechanism**
> We thank the reviewer for this great question.
> As our best-known, mainstream defense mechanisms against our attack include adversarial training, post-correction, and pre-processing.
> First, adversarial training adds adversarial examples to the training data to enhance model robustness. Our attack method could also benefit adversarial training by generating adversarial examples.
> Second, post-correction is already included in our attack setting. Our method could effectively disable the post-correction.
> Finally, pre-processing such as image filtering could remove most of the perturbations, but it would also distort the image quality.
> As for the "adaptive STR model" mentioned by the reviewer, we are not aware of a specific method with this name.
> We would greatly appreciate it if the reviewer could provide more details, which would help us better understand and address this point.
>
>
> **W4/Q1: Question about the CEF.**
> We thank the reviewer for this insightful question.
> First, regarding the coevolution energy function (CEF, Eq. 7), the "global constraint" refers to the information of the best pixels from the other $N-1$ populations. Specifically, we generate a combined perturbation $\Delta(\lbrace p_i^j \rbrace \cup \lbrace p_{k}^* \rbrace_{N-1})$ and evaluate its attack performance to determine the energy of the current attack pixel $p_i^j$.
> Second, our method employs a DE algorithm within each population to search for attack pixel locations. As DE itself cannot guarantee finding the global optimum, our method also cannot ensure a global optimum.
>
> **W5: Application scenarios.**
> We thank the reviewer for this valuable suggestion.
> A specific scenario in privacy protection is: when users upload images to social platforms, we could add tiny perturbation pixels to the images so that the private texts may not be correctly recognized and collected by malicious OCR systems.
> Except for privacy protection, our attack method could potentially be applied in other contexts, such as poisoning training data, misleading OCR systems, or concealing toxic or prohibited content.
> We will further elaborate on these potential scenarios and their implications in the camera-ready version.
>
>
> **Q2: Influence of the partition method.**
> We thank the reviewer for this good question.
> First, when $N=20$, we use 20 attack pixels to generate adversarial examples, which leads to greater distortion than $N=10$ (default setting).
> Second, the partitioning strategy affects the final locations of these attack pixels, which may cause slight variations in distortion—either an increase or a decrease—though the impact is generally minor.
>
> **Q3: Obvious disturbance points.**
> We thank the reviewer for highlighting this concern.
> Our work mainly focuses on searching for the attack pixel locations while the attack pixel values are set to 0 or 255.
> If the values of 0 or 255 can successfully attack the target model, we directly use these values; otherwise, we use DE again to search for other attack pixel values.
> To improve imperceptibility, we are exploring better value-search strategies.
> A simple approach is to start from the original pixel value $v_0$ and iteratively try $v_0 \pm 1$, $v_0 \pm 2$, and so forth until a successful perturbation is found.
>
> **Q4: The population setting.**
> We thank the reviewer for this good question.
> As shown in Table 4, with the same total number of pixels ($N \times M$), a larger $N$ yields better attack performance. Thus, we recommend using a larger $N$ with a smaller $M$. These parameters can also be adjusted automatically based on the text length; for example, if we aim to perturb each character by one pixel, $N$ can be set equal to the number of characters in the image.

---

> > ### Comment · Reviewer_1ydv · 2025-08-07
> >
> > The responses have mostly addressed my concerns. Based on  the feedback from the authors and the other reviewers' comments, I would like to keep my original score.

---

### Official Review · Reviewer_1goM · 2025-07-03

**Clarity:** 3
**Significance:** 2
**Originality:** 3
**Rating:** 4
**Confidence:** 4

**Summary:**

This paper aims to attack the STR model with irreversible output since text string is contextual and less wrong character can be corrected with dictionary or LLMs. They propose the Multi-Population Coevolution Search (MPCS) method to attack each character in the image. They  decompose the global optimization objective into sub-objectives to solve the attack pixel concentration problem existing in previous attack methods. To address the new combined perturbation shift problem, they propose a coevolution energy function and make the theoretical analysis. Experiments on CRNN, ASTER, SAR and IGTR demonstrate the results.

**Questions:**

See "Weaknesses".

**Ethical Concerns:**

["NO or VERY MINOR ethics concerns only"]

**Final Justification:**

See my latest comment.

**Limitations:**

Yes.

**Paper Formatting Concerns:**

N/A.

**Quality:**

2

**Strengths And Weaknesses:**

Strengths:
1) Finding the phenomenon that text string attacked with few wrong characters is easily recovered is valuable since in many scenarios the dictionary is used and the LLMs' abilities progress rapidly.
2) The proposed MPCS performs better in irreversible settings.
3) The theoretical analysis guarantees the superiority of the proposed coevolutionary energy function.

Weaknesses:
1) The proposed method is complex and the SR rate decreases with large margins. SR* is important, but SR is also an important metric to evaluate different methods, the decrease from 90%+ to 80%+ demonstrates that the proposed method's success rate is too low.
2) Equation 5, theta_max is not defined, and is it (255, 255, 255)?
3) Tab.1, the results of SAR+IC13 show the proposed method is ineffective, please explain it in detail.
4) Tab.2 lacks SR* results, please explain.
5) Tab.3, ES gives the negative effect. In my opinion, the attack time overhead is not so important than that of SR.

---

> ### Author Rebuttal · Authors · 2025-07-31
>
> **Q1: Decreased SR.**
> We thank the reviewer for highlighting this concern.
> The decrease in SR mainly stems from two factors: (1) the combined perturbation shift phenomenon, and (2) the early-stop mechanism.
> To address the first, we introduce the CEF, which effectively alleviates its impact and improves SR, as shown in Table 3.
> The early-stop mechanism, on the other hand, is designed to reduce the generation number (Figure 4), representing a deliberate trade-off between SR and attack time.
>
> Besides, we would like to emphasize that our primary goal is to design **irreversible attacks**.
> In STR scenarios, dictionary-based correction is a common post-processing step, so attacks that remain effective after such correction are of greater practical significance.
> Thus, SR* (success rate after correction) and PR (perturbation rate) are more indicative of performance than SR (before correction).
> Importantly, the SR drop after correction (SR $-$ SR*) reflects attack reversibility: our method shows only a slight decrease (0.61\%, 87.36\% $\rightarrow$ 86.75\%), whereas the second-best method suffers a drastic drop (35.51\%, 95.60\% $\rightarrow$ 60.09\%).
>
>
> **Q2: Symbol definition.**
> We thank the reviewer for pointing out this error.
> Indeed, the $\theta_{max}$ in equation 5 is (255, 255, 255).
>
> **Q3: Lower performance.**
> We thank the reviewer for highlighting this concern.
> We believe there are three main factors contributing to the relatively lower performance on SAR+IC13.
> (1) The IC13 dataset consists of easy samples with normal fonts aligned horizontally, which increases the attack difficulty.
> (2) The SAR model is designed for irregular text recognition, which means it is naturally robust against adversarial examples.
> (3) **Most importantly**, OnePixel has no constraint on the attack pixel location, so it could allow many attack pixels to concentrate in a small region and change a single character. That explains its higher SR.
> However, we want to note that our perturbation rate (PR) is still the highest (55.31\% vs. 41.40\%), and the SR drop after correction (SR $-$ SR*) is also smaller than other methods (18.45\% vs. 26.35\%).
> As we explained in **Q1**, these metrics are much more important for evaluating attack irreversibility, and the results demonstrate the best irreversibility of our attack method.
>
> **Q4: Missing metric in Table 2.**
> We thank the reviewer for highlighting this concern.
> We didn't calculate the SR* metric in Table 2 because it is redundant and expensive.
> As shown in Table 2, compared to other attack paradigms except for UDUP, which has extremely low SRs, **our method achieves much higher SR (9.54\% $\sim$ 38.71\%) and PR (27.37\% $\sim$ 67.28\%)**.
> Since a higher SR (before correction) and PR (more incorrect characters) directly imply a higher SR* (after correction), we believe the SR* metric is not necessary in this table.
>
> Due to time constraints, we compute the SR* metric for the second-best method only (Sparse-rs), as a representative comparison.
> On CRNN and CUTE80, Sparse-rs achieves an SR of 63.73\% and a PR of 40.23\%, which are 23.63\% and 54.31\% lower than our results, respectively.
> After applying text correction, its SR* drops to 22.68\%, which is 64.07\% lower than ours.
> These results indicate that a lower PR typically leads to a more significant drop in success rate after correction (i.e., SR - SR*), further supporting the advantage of our method.
>
> **Q5: The ES mechanism.**
> We thank the reviewer for highlighting this concern.
> The ES mechanism provides a trade-off between attack time and SR. If the attackers want the best SR regardless of attack time, they can cancel the ES mechanism by setting the parameter "early_stop" to 0 in our provided code.

---

> > ### Comment · Reviewer_1goM · 2025-08-06
> > **Good motivation and extensive evaluation**
> >
> > The rebuttal has explained the concerns in detail, and I rate this submission as good motivation and extensive evaluation. I finalize my rating as borderline accept.

---

### Official Review · Reviewer_cRqd · 2025-07-03

**Clarity:** 3
**Significance:** 3
**Originality:** 2
**Rating:** 5
**Confidence:** 4

**Summary:**

In this paper the authors present a pixel-level blackbox attacked method targeting  to manipulate scene text images. The authors use  a method named an MPCS to attack each character in the image to avoid being easily corrected by external dictionaries, and shows the effectiveness of their proposed method using four STR datasets.

**Questions:**

1.  The authors mention that  they used early stopping to balance the extra time caused by maximum generation (MG). Is it possible to quantify the average attack time per  sample text-line image with and without early stopping?
2.  I have already mentioned it in the strength and weakness part above.  Since there are many Non-Latin scent text datasets, is there a specific reason for selecting the Latin script? Was it intentional?
3. The authors provide different metrics such as SR/SR*, PR, and L2 distance, is it possible to provide sample real examples and compute the values using each metric,  the examples can be include in the supplemental part?

**Ethical Concerns:**

["NO or VERY MINOR ethics concerns only"]

**Final Justification:**

The questions i raised have been addressed, and the authors also presented a new experiment for testing their method on Non-latin scripts, which I had previously commented on as a weakness. So, I finalize my rating as accept.

**Limitations:**

Yes

**Paper Formatting Concerns:**

The paper is well written and there is no major issue with formatting, but it needs  a space between the caption and table of Tbale 3 to make it consistency with other tables.

**Quality:**

3

**Strengths And Weaknesses:**

**Strengths**:
- The paper is well written and easy to follow.
- The experiment section provides detailed results with clear and concise reasoning for the hyperparameters and the methods chosen.

**Weaknesses**
- The method presented in this paper is evaluated solely on the Latin/English script, and it is unclear whether it can be extended to different scripts or it is script dependent. It would be nice if the authors could test additional Non-Latin scene texts.

---

> ### Author Rebuttal · Authors · 2025-07-31
>
> **Q1: Average attack time with early stop.**
> We thank the reviewer for this good question.
> The attack time varies across different devices.
> On our device with a 3090 GPU, the average attack time on CRNN with the CUTE80 dataset is 18.23 seconds w/. ES and 37.17 seconds w/o. ES.
> This result demonstrates the effectiveness of the ES mechanism.
>
> **Q2: Datasets selection.**
> We thank the reviewer for pointing this out.
> We agree that evaluating on non-Latin scripts would further validate the generality of our method.
> Our current experiments rely on STR models from MMOCR and OpenOCR, which are trained on large-scale English datasets.
> Hence, we have only conducted evaluations on four widely used English datasets.
> Importantly, our method itself does not rely on script-specific assumptions, and we expect it to generalize to non-Latin scripts.
> To validate this, we conducted a set of comparative experiments on a Chinese text recognition benchmark (from “Benchmarking Chinese Text Recognition: Datasets, Baselines, and an Empirical Study”), comparing our method with $\mathrm{AD}^2\mathrm{E}$.
> Due to time constraints, we randomly select 200 images from this benchmark and apply both attacks on the IGTR model, which is trained on Chinese datasets by OpenOCR.
> Our method achieves a success rate (SR) of 51.85\% and a perturbation rate (PR) of 34.44\%, while $\mathrm{AD}^2\mathrm{E}$ achieves 38.89\% SR and 21.12\% PR.
> These results suggest that our method generalizes well to non-Latin scripts.
>
>
> **Q3: Metric calculation example.**
> We thank the reviewer for the suggestion.
>
> (1) The perturbation rate (PR) metric.
>
> \begin{equation}
>     \textbf{PR} = \frac{1}{S}\sum_{i=1}^{S}{\frac{D(Y_i, \mathcal{F}(X_i^{adv}))}{len(Y_i)}},
> \end{equation}
>
> For example, if the ground truth $Y_i$ is "ABCD", and the predicted label $\mathcal{F}(X_i^{adv})$ is "CDAB", then the edit distance between them is $D(Y_i, \mathcal{F}(X_i^{adv}))=2$.
> The length of the ground truth is $len(Y_i)=4$.
> So the perturbation rate of this sample is 50\%.
>
> (2) The success rate before/after correction (SR/SR*).
> For example, if the number of samples is 1000, and 800 of them are successfully attacked, then the SR is 80\%.
> Next, we use an LLM to correct the mispredicted texts.
> If 200 of them cannot be corrected, then the SR* is 20\%.
>
> (3) The L2 distance metric.
> This metric evaluates the average L2 distance between the original image $X_i$ and the adversarial example image $X_i^{adv}$:
> \begin{equation}
>     \textbf{L2} = \frac{1}{S}\sum_{i=1}^{S}||X_i - X_i^{adv}||_2.
> \end{equation}
>
> We will provide such examples in the supplementary material.

---

> > ### Comment · Reviewer_cRqd · 2025-08-06
> > **clarification on non-latin script test and examples**
> >
> > The response addressed the questions I raised, and I trust the L2 metrics example  will be added in the supplementary, but I would like to know the authors plan to present their new non-latin script test to reader.

---

> > > ### Author Response · Authors · 2025-08-06
> > >
> > > We sincerely thank the reviewer for the positive feedback and for acknowledging our responses.
> > > We confirm that we will include the L2 metrics example in the supplementary material.
> > > As for the new non-Latin script experiments, we plan to present the test results in the camera-ready version (in Section 6 or the supplementary material), along with additional visualizations.
> > > This will help clearly demonstrate the generality of our method across different scripts to the readers.

---

> > > > ### Comment · Reviewer_cRqd · 2025-08-06
> > > >
> > > > Thank you for the response and the additional experiment.

---

### Note · Authors · 2025-08-12

We thank the reviewers and the AC for the constructive feedback and valuable discussions during the review process.
Our work proposes a new STR attack objective—maximizing the number of altered characters—to mitigate the effectiveness of common post-processing methods in STR scenarios and achieve irreversible attacks. To achieve this, we introduce a novel Multi-Population Coevolution Search (MPCS) pixel-level black-box attack, which significantly increases misrecognized characters and produces more irreversible attacks.

During the rebuttal, we addressed all raised concerns. We appreciate the reviewers’ acknowledgment of our “good motivation and extensive evaluation” and their suggestions to improve presentation.
We are committed to revising the paper for greater clarity, including clearer metric definitions, expanded non-Latin script evaluation, and supplementary results.

Given the novelty of our approach, the empirical improvements over strong baselines, and the practical implications for improving STR robustness, we believe this work would be of significant value to the community. We respectfully hope the AC and reviewers will consider our responses and the merits of this contribution in making the final decision.

---

### Decision · Program_Chairs · 2025-09-17

**Decision:**

Accept (poster)

**Comment:**

(a) This paper introduces a novel black-box adversarial attack method for Scene Text Recognition (STR) models, named Multi-Population Coevolutionary Search (MPCS). The key contribution of the paper is the focus on "irreversibility” i.e., generating adversarial examples that are resistant to correction by dictionaries or Large Language Models. The method addresses the "pixel concentration" problem of prior attacks by dividing the image into horizontal regions and running parallel evolutionary searches. To solve the subsequent "combined perturbation shift" problem that this decomposition creates, the authors propose a co-evolutionary energy function to ensure the perturbations work in tandem. The paper's findings, validated across multiple STR models and datasets, show that MPCS significantly increases the number of mispredicted characters (Perturbation Rate), resulting in attacks that maintain a high success rate even after post-correction attempts.

(b) around the concept of "irreversibility," which reviewers found to be a valuable and significant contribution to the field. The proposed MPCS framework is well-motivated, systematically designed, and supported by strong theoretical analysis, particularly for the innovative co-evolutionary energy function that addresses a key challenge in multi-agent optimization. The experimental evaluation is another major strength, praised by reviewers for its thoroughness, with tests conducted on multiple standard datasets against a wide range of STR models and baseline attack methods, all of which clearly demonstrate the superiority of the proposed approach.

(c) The most significant weakness noted by reviewers was that the method was evaluated solely on English/Latin script, which raised questions about its generalizability to other languages. Some reviewers also found the method to be complex, raising concerns about the computational cost and the trade-off between the attack success rate (SR) and the more crucial irreversible success rate (SR*). Minor weaknesses included a lack of clarity on certain concepts like the "combined perturbation shift problem" and questions regarding the choice of baselines and specific experimental settings which was addressed by the authors in the rebuttal.

(d) The recommendation is to Accept this paper. The primary reason is that it introduces a novel and practically important concept—"irreversibility"—to the domain of adversarial attacks on STR models and provides a technically solid and well-validated method to achieve it. The authors' rebuttal was strong, directly addressing the main weakness by conducting a new experiment on a non-Latin (Chinese) dataset, which successfully demonstrated the method's generalizability and addressed the concern of reviewer cRqd. The final reviewer consensus is mostly positive with the only negative review coming from reviewer HCsA who has not been involved in the reviewal process despite few reachouts. The paper provides a clear contribution, advances the state-of-the-art in a meaningful way, and has review consensus and I am recommending a poster based on the above.

(e) The rebuttal period was well managed by authors mostly and demonstrated the authors' commitment to addressing reviewer feedback. The most critical point raised by Reviewer cRqd was the evaluation's limitation to Latin scripts. The authors directly addressed this by conducting a new experiment on a Chinese text recognition benchmark, showing that their method generalized well and outperformed a key baseline. This new evidence led the reviewer to finalize an "Accept" rating. Other concerns from Reviewers 1goM and 1ydv regarding performance trade-offs (SR vs. SR*), the method's complexity, and its performance on long texts were also thoroughly addressed with detailed explanations and supplementary experiments. The authors successfully clarified that the slight decrease in the standard success rate (SR) is a deliberate trade-off to achieve a much higher irreversible success rate (SR*), which is the paper's primary goal. This active and successful engagement solidified the positive opinions of the reviewers. Unfortunately, reviewer HCsA has not addressed the author’s rebuttal even though the authors provide responses to their key concerns.